# Effect of Growth Hormone and Estrogen Replacement Therapy on Bone Mineral Density in Women with Turner Syndrome: A Meta-Analysis and Systematic Review

**DOI:** 10.3390/ph16091320

**Published:** 2023-09-19

**Authors:** Weronika Szybiak, Barbara Kujawa, Miłosz Miedziaszczyk, Katarzyna Lacka

**Affiliations:** 1Students’ Scientific Section at the Department of Endocrinology, Metabolism and Internal Medicine, Poznan University of Medical Science, 60-355 Poznan, Poland; weronikaszybiak@gmail.com (W.S.); kujawabasia@interia.pl (B.K.); 2Department of Nephrology, Transplantology and Internal Medicine, Poznan University of Medical Science, 60-355 Poznan, Poland; m.miedziaszczyk@wp.pl; 3Department of Endocrinology, Metabolism and Internal Medicine, Poznan University of Medical Science, 60-355 Poznan, Poland

**Keywords:** Turner syndrome, bone mineral density, growth hormone, estrogen replacement therapy, pharmacotherapy

## Abstract

Osteoporosis is a serious implication of Turner syndrome (TS). Common methods for the treatment of TS are growth hormone (GHT) and estrogen replacement therapy (ERT). We examined the relationship between the treatment of TS and bone mineral density (BMD) of the lumbar spine. The purpose of our study was to show the currency of BMD states among patients with TS for treatment with GHT and ERT. We searched databases for studies published from inception to April 2023. The articles were related to TS, osteoporosis, ERT, GHT, BMD and treatment patients with TS. We applied the selection criteria: lumbar spine values at L1–L4; dual-energy X-ray absorptiometry (DXA); treatment which was applied: one group of articles: ERT and two group of articles: GHT; results performed as means ± SD. In total, 79 articles were analyzed, of which 20 studies were included and 5 were considered for meta-analysis. The total number of women in the articles selected was 71. Based on the results of the meta-analysis, the effect of ERT on BMD demonstrated a significant increase in BMD (the standardized mean difference in the random model was 0.593 g/cm^2^, 95% CI: 0.0705 to 1.116; *p* = 0.026), which showed that treatment with estrogen particularly increases bone mass during treatment, which contributes to reducing the risk of fractures. The effect of GHT on BMD demonstrated a non-significant decrease in BMD in patients with TS. The results for growth hormone show that this therapy does not improve bone density. However, our review emphasizes the beneficial effect of supplementing growth hormone (GH) on the clinical presentation of TS.

## 1. Introduction

Turner syndrome is a congenital disease caused by aberrations in one of the sex chromosomes (X) [1]. TS occurs in approximately 1 of 2500 alive female births. The key signs of untreated women with TS are: stunted growth, lack of sexual development (primary amenorrhea, sexual infantilism due to hypergonadotropic hypogonadism) and numerous congenital anomalies of which the most frequent are: changes in the skeleton, palpebral ptosis, lymphedema, webbing of the neck, cardiovascular anomalies such as hypertension, congenital heart diseases (coarctation of the aorta, subaortic stenosis, ventricular septal defect) and renal anomalies (i.e., horseshoe kidney, double pelvis and double kidney). Other signs of TS include: low hairline (40%), gothic palate (40%) or nail defects (60%), eye anomalies and a tendency for obesity [2,3,4]. Some of the above-mentioned symptoms may also occur independently and not be a symptom of Turner syndrome at all, but only a single mutation or appearance feature. An example of a study from 2019 shows that there may also be cases of supernumerary teeth among patients with TS. Hyperdontia is an anomaly characterized by an excessive number of teeth, both erupted and unerupted. It most often occurs occasionally but can also be associated with genetic syndromes such as Gardner’s syndrome, Down’s syndrome, Noonan’s syndrome, Fabry–Anderson disease and cleidocranial dysplasia. In these cases, it is a syndromic form associated with a complex clinical picture of the disease. Therefore, it is worth differentiating individual features of the patient: whether they belong to the picture of the disease or are only similar non-syndromic conditions [5,6].

Turner’s syndrome is diagnosed by cytogenetic examination and can take various forms: monosomy of the X chromosome (45,X), mosaic-type karyotype (e.g., 45,X/46,XX), and structural aberration (partial or complete deletion of the short arm of the X chromosome (delXp); complete deletion of the long arm of the X chromosome (delXq); isochromosome in the long arm of the X chromosome (i(Xq)); marker chromosome) [2,7].

In Turner syndrome, there are also characteristic skeletal abnormalities, which include deformity of medial condyle of the tibia, Modelung’s deformity, numerous changes in the hands’ fingers, the metacarpal signs. Short stature is also characteristic of women with Turner syndrome, which results from the haploinsufficiency of the pseudoautosomal gene SHOX. Patients with Turner syndrome are not deficient in growth hormone, but treatment with GH is generally used and leads to an increase in final height of about 7 cm [1].

Decreased BMD and osteoporosis among patients with Turner syndrome develop much earlier than postmenopausal osteoporosis. It is associated with ovarian failure and an increased risk of bone fractures, especially in childhood and after the age of 45.

Currently, osteoporosis is considered to be one of the most common diseases affecting bones in the world. It is defined as a disease in the image of decreased bone mineral density (BMD), reduced bone mass and microarchitectural deterioration of bone tissue. Aggravation of the status of bone tissue concludes in a decline in bone strength and an increased prevalence of fractures [8,9]. Increased bone fragility is attributed to the direct or indirect impact of X chromosome abnormalities, estrogen deficiency as well as comorbidities with TS (pro-inflammatory cytokines associated with obesity or inflammatory bowel disease, elevated FSH levels) [2,10]. Bone malformation, impaired balance and visuospatial cognitive dysfunction can occur, leading to a propensity to falls, fractures and lower quality of life. Bone abnormalities and associated fractures also depend on environmental factors such as physical activity and the patient’s muscular strength. An active lifestyle and regular physical activity can have a positive effect on bone health among these patients [11].

The primary treatment for Turner syndrome involves the supply of GH and ERT. While GHT and ERT are separate treatment modalities, they often work synergistically to optimize the growth potential and overall health of individuals with Turner syndrome. First, GHT is implemented and focuses on achieving height by girls that will not constitute physical limitations and starting sexual maturation at an age similar to their peers. Estrogen therapy is usually initiated when growth hormone therapy has resulted in sufficient linear growth and the individual enters adolescence. The introduction of estrogen at this stage promotes the development of secondary sexual characteristics and helps achieve normal pubertal milestones. According to the 2016 Cincinnati International Turner Syndrome Meeting, treatment with GH should start at around 4–6 years of age, and no later than about 12–13 years old, especially when the child already has evidence of growth failure or short predicted adult height. The recommended basic GH dose is 45–50 µg/kg/day or 1.3–1.5 mg/m^2^/day (4.0–4.5 IU/m^2^/day). The recommended age at which to start ERT is 11–12 years of age, increasing to an adult dosage about 2–3 years later. Therapy is continued until little growth potential remains (bone age ≥ 14 years and HV < 2 cm/year) or until the girl maintains a height which is satisfying for her. Estrogen replacement options include: transdermal estradiol (E2) (3–7 µg/day for pubertal initiation, 25–100 µg/day for adult), micronized 17β oral E2 (0.25 mg/day for pubertal initiation, 1–4 mg/day for adult), ethinyl estradiol (EE) (2 µg/day for pubertal initiation, 10–20 µg/day for adult), and depot E2 (0.2 mg/month for pubertal initiation, 2 mg/month for adult). ERT is discontinued after reaching the typical age of menopause, until the risk of complications is higher than advantages of continuing the treatment [12]. Both therapies affect bone changes in the treated patients. ERT is one of the modifiable factors which improve bone status and prevent longitudinal consequences of osteoporosis [13]. Estrogen influences decreased bone resorption and is crucial to obtain the maximal peak bone mass [10]. GHT helps achieve longitudinal bone growth and induce subperiosteal bone formation which affects the increase in cortical bone mass. The results of research conducted on rats treated with GH show that this therapy has a beneficial impact on the increase in bone volume, mineralization surface, osteoid surface and osteoclastic surface [13]. However, human studies show that the influence of GHT on BMD is ambiguous [1]. Patients who are not treated with estrogens and growth hormone show increased susceptibility to bone fractures (45%) [2].

The aim of our study was to conduct a meta-analysis according to PRISMA guidelines in order to evaluate and compare the effect of human growth hormone and estrogen therapy on bone mineral density (BMD) in patients with Turner syndrome.

## 2. Results

Data from selected articles were analyzed and compiled in Table 1 and Table 2. Table 1 collects the data of patients treated with ERT, taking into account differences between populations in age, height, karyotype and BMD but also differences in age of start ERT and duration of treatment. The percentage of patients treated additionally with GHT was also taken into account.

Table 2 compares the data of patients treated with GHT and without this treatment. It takes into account differences in age, height and BMD between patients with and without GHT. Additionally, the duration of treatment was included in populations treated with GHT. The percentage of patients treated additionally with EHT was also taken into account.

Statistical analysis and the attached figures were performed using MedCalc software, version 20.006.

The studies containing sufficient data on the effect of ERT on BMD demonstrated a significant increase in lumbar spine areal BMD in patients with Turner syndrome. The standardized mean difference in the random model was 0.593 g/cm^2^ (95% CI: 0.0705 to 1.116; *p* = 0.026). The heterogeneity between studies was I^2^ = 56.89% (95% Cl for I^2^ = 0.00 to 87.71; *p* = 0.0983). The results of Egger’s test show that there is a statistically significant publication load. Egger’s test −9.8452 (95% Cl = −29.1591 to 9.2944; *p* = 0.0962). The results are presented in Figure 1.

The studies with sufficient data examining the effect of GHT on BMD demonstrated a non-significant decrease in lumbar spine areal BMD in patients with Turner syndrome. The standardized mean difference in the random model was −0.028 g/cm^2^ (95% CI: −0.359 to 0.303; *p* = 0.868). The heterogeneity between studies was I^2^ = 0.00% (95% Cl for I^2^ = 0.00 to 93.73; *p* = 0.5855). The results of Egger’s test show that there is no statistically significant publication load. Egger’s test 2.9326 (95% Cl = −24.8244 to 30.6897; *p* = 0.4076). The results are presented in Figure 2.

The meta-analysis indicates that ERT has an effect on BMD in the lumbar spine among women suffering from TS with a statistical significance of *p* < 0.05. In turn, the results of the influence of GHT on BMD in the lumbar spine among patients with TS do not show statistical significance for these results (*p* > 0.05). 

## 3. Discussion

Processes of osteogenesis, osteoclastogenesis and regeneration of skeletal tissue are multistage and composite. Specialized cells such as osteoblasts, osteoclasts and osteocytes are involved in these complicated processes. The balance between the activities of these cells is especially important for the proper growth, development, stabilization and mineralization of bone tissue [19,20,21]. Under normal conditions, the bone resorption process driven by osteoclasts is in balance with the production of the osteoid by osteoblasts. Both processes determine the correct bone turnover [22,23,24].

The activity and interactions of the three types of bone cells are regulated by hormonal, molecular and genetic factors. The effect of estrogen on bones provides a balance between bone resorption and osteogenesis [19,25,26]. Research has shown that in long-term ERT, estrogen promotes the differentiation of osteoblasts and inhibits adipocyte differentiation in bone marrow stromal cell lines [27].

Osteoblasts are capable of regulating osteoclast function by secretion of cytokines and growth factors. Estrogen’s function is to control the number of cytokines and growth factors, so it indirectly supervises osteoclast formation [17,28]. Moreover, research conducted by Boyce with in vivo treatment of 17-beta estradiol showed that estrogen promotes apoptosis in osteoclasts and inhibits apoptosis in osteoblasts and osteocytes [29]. Estrogen also decreases the secretion of cathepsin B, cathepsin L and thyroid hormone receptor-associated protein (TRAP), which entails a decreased secretion of lysosomal enzymes [30].

The function of estrogen is to regulate the expression of hormones that affect bone metabolism: promote calcitonin secretion and increase the level of 1,25-(OH)2D3 (by increased activity of 25-hydroxylase and renal 1 alpha-hydroxylase). In addition, estrogen can inhibit the secretion of parathormone (PTH) by reducing the point at which PTH reacts to blood calcium [17]. Malfunctions of regulatory factors can lead to abnormal bone structure or mineralization and result in the development of osteoporosis, fractures or deformities [31]. 

Different factors which influence bone metabolism are frequently impaired in patients with TS. Clinical diagnosis of osteoporosis is mainly based on BMD measurement in the lumbar spine, total hip or femoral neck and reported T-scores [32]. The basic methods of treating bone disorders in TS are ERT and GHT. A new and promising approach to bone tissue regeneration is platelet rich plasma (PRP). A study by F. Inchingolo et al. showed that the use of PRP as a platelet concentrate in combination with bone, bone material and bone substitutes showed a beneficial effect on the process of promoting maxillary and mandibular bone healing. So far, however, this method is still controversial and there is insufficient evidence to confirm its effectiveness. However, there are hopes that in the future, it may be widely used in surgery [33].

Individual factors affecting bone metabolism and structure in TS patients are described in detail below.

### 3.1. Correlation between Estrogen and BMD in Turner Syndrome

Women with TS suffer from ovarian insufficiency, which leads to chronically decreased levels of estrogen. Estrogens play a significant role when it comes to the suppression of bone resorption and can even have an anabolic effect [11]. Estrogen binds with special receptors on osteoblasts and osteoclasts and influences their activity. Girls with TS suffer from a lack of estrogenic function of the gonads, resulting in the absence of a pubertal growth spike and an increase in bone mineralization before and after menarche. Hypoestrogenism can be corrected by the implementation of ERT. Exogenous estrogen contributes to the optimal peak bone mass and prevents osteoporosis [34]. 

Research on the impact of estrogen on BMD was carried out using DXA. Lumbar vertebrae (L2–L4) were examined, and the results were compared, taking into account the age of start and duration of ERT. Particular studies revealed that using ERT significantly increases BMD in patients with TS and earlier initiation of using estrogen is more effective [16,35].

The purpose of this study was to take into account data from various studies and reveal an average effect of using ERT in patients suffering from TS. The analysis also confirmed the beneficial impact of ERT on areal lumbar spine BMD in TS patients, which was consistent with the results of individual studies. The results of individual studies are represented in Figure 3. Statistical analysis demonstrated an average increase in BMD in L1–L4 localization in patients who apply ERT. The standardized mean difference in the random model was 0.593 g/cm^2^ (95% CI: 0.0705 to 1.116; *p* = 0.026). Our results correspond with previous results of analysis investigated by Cintron et al. Their study also confirmed the positive influence of ERT on BMD in women with TS, but their inclusion criteria take into account age under 40 [36]. We resolved not to apply age as an inclusion criterion because the aim was to determine the effect of ERT on BMD in the overall patient population. According to the clinical practice guidelines, ERT should start between 11 and 12 years of age, and it is also recommended to include progesterone after two years after starting ERT [12].

### 3.2. Polymorphisms of Estrogen Receptor and Effectiveness of ERT

The effectiveness of ERT can be correlated not only with the dose and time of use but also with the individual’s genetically determined response to therapy [37]. Estrogens have different effects on osteoblasts and osteoclasts depending on the genetic polymorphisms of estrogen receptors (ER). This difference is due to the varying binding and affinity of estradiol to polymorphic receptors [17,34].

Sowińska-Przepiera et al. showed a relationship between a polymorphism in the ER-alpha gene and the effectiveness of ERT on the increase in BMD in girls with TS. The correlation between PvuII and Xbal ER-alpha polymorphism and BMD in TS patients was analyzed. The study showed a remarkable increase in BMD after four years of ERT in TS patients with subsequent genotypes: xx and Xx of the Xbal gene and pp and Pp of the PvuII gene. On the other hand, women with XX (Xbal gene) and PP (PvuII) genotypes were characterized by a poorer response to estrogen treatment and lower BMD values. The researchers also suggested that including bisphosphonates in the treatment of TS patients with the XXPP genotype could help improve their BMD [36]. 

Polymorphisms were not taken into account in our meta-analysis due to the lack of insufficient data, which can be seen as a limitation of our findings. Knowledge about the correlation between polymorphisms and the effectiveness of ERT can be significant in defining the proper dose and time to start ERT to achieve eligible results for BMD in patients with TS [36].

### 3.3. Polymorphisms of Growth Hormone Receptor and Effectiveness of GHT

Response to GHT depends on many factors, including age of initiation of therapy, diet, dose, frequency and duration of therapy. This may also be influenced by genetic variability directly related to the growth hormone receptor (GHR) [37,38]. The GHR is composed of 3 regions encoded by a gene located on chromosome 7. Exons 3–7 encode the extracellular region, of which the haplotype: fl/fl (presence of exon 3) results in a full-length receptor (GHR fl-fl or wild type). The shorter form of the receptor (GHR fl-d3 and GHR d3-d3) is associated with a lack of exon 3, which may affect changes in receptor stability, transport and processing [39]. 

In their study, Dos Santos et al. showed that fibroblasts transfected with the GHR d3 allele have a higher level of GHR signaling after administration of exogenous GH. During 1 year of GHT observation, they showed that the response to GH is significantly better among children carrying 1 or 2 copies of the GHR d3 allele compared to those homozygous for the full-length allele (GHR fl-fl) [40].

These results were confirmed in 2009 by Wassenaar et al. conducting a meta-analysis, which showed that the GHR d3 genotype causes faster growth during 1 year of treatment (about 0.5 cm) and treatment is more effective with the supply of low doses of GH [41]. Similar results were obtained in 2012 by Renehana et al., however, they did not show a modifying effect of age on the use of therapy in the assessment of the GHR d3 allele [42].

### 3.4. Correlation between GHT and BMD in TS

Patients with TS are also treated with GHT to increase their height in adult life. GH affects bones directly and indirectly for osteoblasts and osteoclasts. Acting on osteoblasts increases proliferation and stimulates differentiation which is measured by an increased production of osteocalcin, alkaline phosphatase and type 1 collagen. The effect of GH on osteoclasts is stimulation or inhibition of recruitment precursors to osteoclasts. GH may also stimulate insulin-like growth factor 1 (IGF-1) production. IGF-1 has an anabolic effect on osteoblasts—increases the cell number and stimulates bone matrix production. To sum up, GH and IGF-1 stimulate bone turnover [43,44,45,46]. Apart from increased height, GHT has other positive effects, namely, decreased adiposity and an increased volume of muscle among girls with TS. Although the positive effect of GHT on the height of women with TS is well proven, the influence on BMD is not clear [1]. Comparing different studies, the results are not unequivocal. 

Individual studies differed in the dose and time of using GHT. Using a higher dose and a longer duration of therapy gave an improvement in BMD in patients with TS [47]. Subsequent research applied therapy with a lower dose and took the BMD measurements. The results obtained were corrected based on a change in bone size. After mathematical analysis, no significant differences between the test and control groups were found [1]. 

Our meta-analysis aimed to collect data from individual studies. The results of individual studies are represented in Figure 4. The inclusion criteria for patients with TS used GHT and measurements of BMD at the lumbar spine by DXA. Our statistical analysis revealed no significant increases in BMD in the case of patients applying GHT. The limitation of our results was the small number of articles that met the inclusion criteria. It seems that the influence of GHT on BMD requires further research.

### 3.5. Genetic Influence 

The occurrence of osteoporosis, visuospatial cognitive dysfunction and disturbances in the organs of hearing and balance lead to an increased risk of fractures [35]. This seems to be approximately 25% higher than in the control group. The most prevalent sites of fractures are the metacarpal bones, femoral neck, lower spine and forearm [4]. Moreover, loss of bone mass is not homogeneous. There is a noticeable defect in the cancellous bone, but the mass of the trabecular bone does not change [48]. 

Furthermore, the way BMD develops in the TS population is still the subject of research. Previous studies showed no remarkable difference in body composition or body mineral status between patients with X0 karyotype and mosaicism [35,49]. According to clinical practice, TS symptoms are most pronounced in patients with X-chromosome monosomy. In women with mosaic-type karyotypes, even spontaneous menstruation can occur [2]. Differences in karyotype in the patients included in our meta-analysis are shown in Table 1 with a particular emphasis on the percentage of monosomy in the group of patients treated with ERT. 

Researchers also emphasize the significant role of the short-stature homeobox-containing gene (SHOX) in bone development [48,49]. This gene is located on the X p terminal, the pseudoautosomal region PAR 1 of the X chromosome. In TS, it results in SHOX haploinsufficiency, which could influence different abnormalities such as Madelung’s deformity, scoliosis or micrognathia [48].

### 3.6. Advantages and Disadvantages of GHT

An important problem that concerns the treatment of GH is, certainly, the body’s response to its use. It is difficult to predict how GHT will affect a given patient, which determines the subsequent results of treatment, including the final height. GHT also carries the risk of side effects resulting from its influence on metabolic processes in the body, e.g., impaired glucose tolerance, production of growth-attenuating antibodies and hyperlipidemia. The effectiveness of therapy depends on factors related to the patient. Patients do not always comply with GH therapy, which may be influenced by, among other factors, injections not given by a child, lower socio-economic status, poorer level of understanding about treatment, duration of treatment. The treatment method of daily injections of the hormone can also be a burden for patients.

The positive effect of GHT on bone remodeling is visible after increasing the concentration of bone markers. This therapy improves bone strength and reduces fracture rates [50]. A definite advantage of using GHT is that patients achieve the height of their peers, which certainly provides them with psychological comfort and prevents stigmatization by society. The risk of the above-mentioned side effects associated with treatment is usually very low and the treatment may contribute to increased health-related quality of life (HRQOL). Proper education, the use of injection pens and public support are very important in achieving the treatment goal [51,52].

### 3.7. Advantages and Disadvantages of ERT

Among the disadvantages of using ERT, one can mention its effect on metabolic changes in the body. Theoretically, estrogen can affect lipids and binding proteins, however, conducted studies did not show significant changes in the lipid profile of treated patients. Similar results could have been obtained from studies examining the effect of estrogens on carbohydrates. ERT may also adversely affect the cardiovascular risk of patients, especially among patients with existing risk factors, such as obesity. The risk is also higher with oral estrogen compared to transdermal patches.

ERT initiated in girls with TS affects sexual maturation: breast development and the appearance of menarche. Estrogens have a positive effect on bone density and thus prevent fractures. In addition, in adolescents with TS, oral estrogen therapy improved self-esteem and psychological well-being. ERT also affected motor speed and verbal and nonverbal processing among teenagers, but there was no improvement in these areas among adults with TS receiving ERT [53]. 

### 3.8. Side Effects of GHT and ERT Treatment

The benefits of hormone treatment often outweigh the risks, especially when under appropriate medical supervision. It is important to discuss any concerns or side effects to ensure the best possible outcomes. 

### 3.9. GHT

The potential side effects that may occur during treatment with the growth hormone include increased blood sugar levels and inducing insulin resistance. The diagnosis of type 1 diabetes occurs among children treated with growth hormone of a similar age where it is diagnosed in the general population. To reduce the risk of diabetes, one should regularly test their blood sugar, exercise and eat a balanced diet. A rare and self-limiting side effect of the therapy is gynecomastia. The time from the start of GH treatment to the appearance of gynecomastia can range from 0.5 months to 8 years. For children complaining of unilateral or bilateral hip or knee pain, slipped capital femoral epiphysis (SCFE) should be considered. Treatment is always surgical. GH and insulin-like growth factor-1 (IGF-1) have mitogenic and anti-apoptotic properties, which is associated with the suspicion that they can induce tumor formation. Although this has not been conclusively proven, it is important to monitor serum IGF-1 levels in patients to ensure that they do not exceed normal values for age and gender. Patients complaining of headaches, nausea, vomiting and symptoms suggestive of papilloedema on fundus examination may suspect Benign Intracranial Hypertension (BIH) or Pseudotumor cerebri. This is the result of the antidiuretic effect of GH. Its presence is confirmed by clinical and imaging tests. If the diagnosis is confirmed, GH should be discontinued. Patients with TS have an increased risk of developing scoliosis and worsening of the existing scoliosis as a result of GH treatment. They should be under the constant care of an orthopedic surgeon and orthopedic evaluation before initiating GH therapy. Other side effects may directly affect the musculoskeletal system and cause muscle and joint pain as well as swelling. A balanced diet, low-impact physical activity and proper hydration can help prevent or reduce symptoms [54].

### 3.10. ERT

The most common complication of ERT treatment among women with Turner syndrome is breast tenderness or swelling. They may experience vaginal bleeding or spotting, especially in the first few months of ERT. Unusual or heavy bleeding should be reported to the doctor by the patient. Mood changes such as mood swings and irritability are also common. They can be alleviated through regular exercise and a healthy lifestyle. Rare but dangerous side effects of ERT include an increased thromboembolic risk. Testing for thromboembolic risk by measuring factor V Leiden and prothrombinase levels should be performed in girls with a personal or family history of Venous Thromboembolism (VTE). The length of ERT treatment is also determined by the possibility of breast cancer. It is thought that girls with TS may have a lower risk because of lower estrogen exposure overall. Regular breast examinations and mammography help detect cancer early and start treatment as early as possible. The effect on blood pressure may vary depending on the type of estrogen you take. Estradiol (E2) replacement therapy lowers blood pressure and causes salt and water retention. This is a completely different action than in the case of the supply of ethinyl estradiol (EE), which causes an increase in blood pressure. In the case of water retention in the body, physical activity and a balanced diet can help [53].

### 3.11. Vitamin D in Turner Syndrome 

In a previous study, some polymorphisms which have an impact on BMD have been distinguished. Vitamin D receptor (VDR) gene polymorphisms seem to have an important influence on spinal BMD in patients with TS [10,55]. A study carried out by López et al. revealed that the Bsm I and Fok I polymorphic sites of the VDR gene can be highly connected with BMD in patients with TS. The lowest lumbar spine BMD occur in patients with genotype bb (in Bsm I site) and ff (in Fok I site), compared to TS women with another genotype. It has also been suggested that the ff genotype is responsible for resistance to vitamin D. Detection of this polymorphism could allow to predict low BMD and severe osteopenia in TS women [48]. Barrientos-Rios et al. revealed the influence of the single-nucleotide variant on phenotype TS-patients. There is a connection between lower BMD and the rs4646536 variant of the CYP27B1. It is the gene for enzyme 1-alpha-hydroxylase which takes part in reactions of transformation of vitamin D to the active form—1α,25-dihydroxyvitamin D3 (calcitriol, 1,25(OH)2D3) [56]. CYP27B1 has significant impact on bone mineralization and metabolism, so it shows influence on the TS-characteristic phenotype. The same study also revealed gene–gene interaction between variants of KL, CYP27B1 and VDR in TS-patients, which also have a significant matter for BMD [57]. BMD results were independent of the use of GHT and ERT, which indicates that the problem of osteopenia and osteoporosis in TS patients is a multifactorial and complex problem. Moreover, the results of previous studies and our meta-analysis indicate that both hormonal treatment and vitamin D supplementation used simultaneously may improve BMD results by various mechanisms. 

This study may have some limitations. The model effect estimates are based on prospective observational studies. As a result, these factors are susceptible to errors and biases that may have affected our model estimates. However, the effects we have obtained are consistent with the results of the studies conducted so far. Our calculations may be conservative and potentially underestimate the total health benefits. Our meta-analysis is based only on the analysis of the effects of ERT and GHT on lumbar spine BMD. In addition, the test may not adequately control for confounding variables that may affect BMD, such as age, weight, nutritional status, physical activity level and other medical conditions. Future studies could include BMD measurements of other parts of the body, however, according to a study conducted by Seok et al., the lumbar BMD study best illustrates the effectiveness of treatment among patients with TS in the target age group. The problem may also be a limited sample size, which can be solved by the necessary collection of further data on ERT and HGT treatment on patients with TS.

## 4. Materials and Methods

### 4.1. Database Search 

The following databases were used to obtain information: PubMed, Scopus, Google Scholar and Medline. These databases were chosen for this review with meta-analysis because they are an important source of comprehensive medical knowledge. The search was conducted with the use of the terms: “Turner Syndrome”, “Turner Syndrome and BMD”, “Growth hormone therapy in patients with Turner Syndrome”, “Estrogen replacement therapy in patients with Turner Syndrome”, “Hormonal replacement therapy in Turner Syndrome” and “karyotype in Turner Syndrome and BMD”. The screening was also conducted without a time limit—from inception to April 2023—because even the oldest articles were important for obtaining as much information as possible. The country of origin and form of the articles were irrelevant. A total of 11,994 articles were found. Study protocol has been registered. ID CRD42023455058. The PRISMA flowchart is presented in Figure 5. 

### 4.2. Study Selection

The selection strategy was to choose articles related to the general information about TS, osteoporosis-related to TS, BMD in patients with TS and the influence of treatment on BMD. 

Articles necessary to create the systematic review were obtained in the full-text version. The next step was to select studies that tested the influence of GHT or ERT on BMD. An analysis of the data was conducted. We selected 79 articles that contained the necessary data. To include articles in the meta-analysis, we decided to apply the following selection criteria: the same section of the skeleton was examined—lumbar spine values at L1–L4;using the same method of examination—dual-energy X-ray absorptiometry (DXA);the same treatment was applied: 1 group of articles: ERT and 2 group of articles: GHT;results performed as means ± SD.

The first selection of studies was based on the title and keywords. Those articles which contain the terms connected with Turner syndrome, ERT, GHT and BMD were included in the next part of the screening. Subsequently, the abstracts of these articles were analyzed to determine eligibility for inclusion. The full-text version of potentially important studies was assessed for inclusion. 

We chose studies in which the amount and unit of data allows us to present the results in the form of a meta-analysis. We excluded studies with differing units of measurement, and other ways of data analysis and also those that concerned non-overlapping age ranges. In our study, we tried to show the impact of ERT and GHT on BMD in patients with TS based on data obtained from original articles. Our study focused on lumbar spine areal BMD which was examined with the use of DXA. However, the limitations of this method should be taken into account. One of the disadvantages is the influence of body size on areal BMD interpretation, which is important in the case of TS patients [14]. Therefore, one of the studies selected for our meta-analysis estimated another parameter—bone mineral apparent density (BMAD). The protocol of estimation BMAD consists of mathematical calculations taking into account areal BMD and the square root of the bone area [58]. Because other articles did not include BMAD in their analysis, it could not be taken into account in our study. The solution to diagnostic difficulties seems to be high-resolution peripheral quantitative computed tomography (HR-pQCT), which allows for more precise bone diagnostics and enables 3D resolution, low level of radiation and assessment of volumetric bone [17,59]. Unfortunately, it is less frequently used than DXA. Due to the greater popularity of DXA, we decided to include it as one of the criteria for inclusion in our meta-analysis.

We decided to compare results obtained from the lumbar spine in patients with TS due to the occurrence of cancellous bone in this area which is useful for assessing the effectiveness of treatment. Moreover, this localization is characterized by greater accuracy for recreating a posture of patients in contrast to the hip joint area. Assessment of BMD based on hip measurements may be more accurate in older patients due to degenerative changes and calcification of the vertebral arteries [15], but in the studies included in the meta-analysis, the oldest age of patients was 58 years. Moreover, the study conducted by Seok et al. concludes that lumbar spine BMD could be more appropriate for assessment of fracture risk of vertebrae, which is one of the most characteristic clinical features of osteoporosis [60].

### 4.3. Risk of Bias Assessment Method

The bias assessment was conducted using the RoB 2 tool for each individual study, by addressing signaling questions in order to determine the risk of bias. Two researchers were involved in this process. The sources obtained for bias assessment included articles and study protocols, if available. The risk of bias will be used to drawing conclusions from the accessed data. The final conclusion will be on based on studies graded “low”. Figure 6 and Figure 7 show the bias assessment. The study by Li L. (2019) was excluded from the meta-analysis due to the high risk of bias.

### 4.4. Statistical Analysis

The meta-analysis was performed using the Hedges g statistic with the fixed effects model. Hedges g is a measure of standardized mean difference. Egger’s test was used to assess publication bias. Evaluation of heterogeneity was performed using the I^2^ statistic.

## 5. Conclusions

The results of the meta-analysis indicate that ERT has a real impact on lumbar spine areal BMD in patients with TS. According to the literature, ERT can have an anabolic effect on patients’ bones. Early initiation of ERT can prevent bone complications of TS, such as fractures. Studies have shown that GHT contributes to gaining greater height in girls with TS. Our meta-analysis indicates the non-significant effect of GHT on lumbar spine areal BMD in TS patients, which is consistent with previous articles. However, despite the lack of proven effect of GHT on BMD, it should be emphasized that GHT has many positive effects on the therapeutic process of patients with TS.

The results of this statistical analysis indicate the effectiveness of hormonal treatment on bone mass and density because this article collects and summarizes data from different countries. Considering this meta-analysis from a clinical perspective, we may conclude that we should focus more on ERT treatment in order to achieve BMD within the normal range. ERT is designed to induce puberty and mimic the hormonal changes that occur during natural puberty in people with Turner syndrome. This includes a gradual increase in estrogen levels, which plays a key role in the development of secondary sex characteristics and bone growth. Adequate estrogen levels help prevent bone loss and reduce the risk of osteoporosis. ERT is usually initiated around the age of natural puberty, and this time is critical to maximizing bone density and preventing long-term bone health problems. Growth hormone therapy is given at an earlier age and is primarily focused on increasing height, but may not have the same direct effect on bone mineralization. When analyzing the articles selected by us, other authors also did not clearly define growth hormone treatment as the most effective for increasing BMD. Our results emphasize the significant impact of ERT on BMD in women with TS, which prevents osteopenia, osteoporosis and bone damage. An increased level of BMD results in a lower risk of fractures, increased motor skills and better patient quality of life.

## Figures and Tables

**Figure 1 pharmaceuticals-16-01320-f001:**
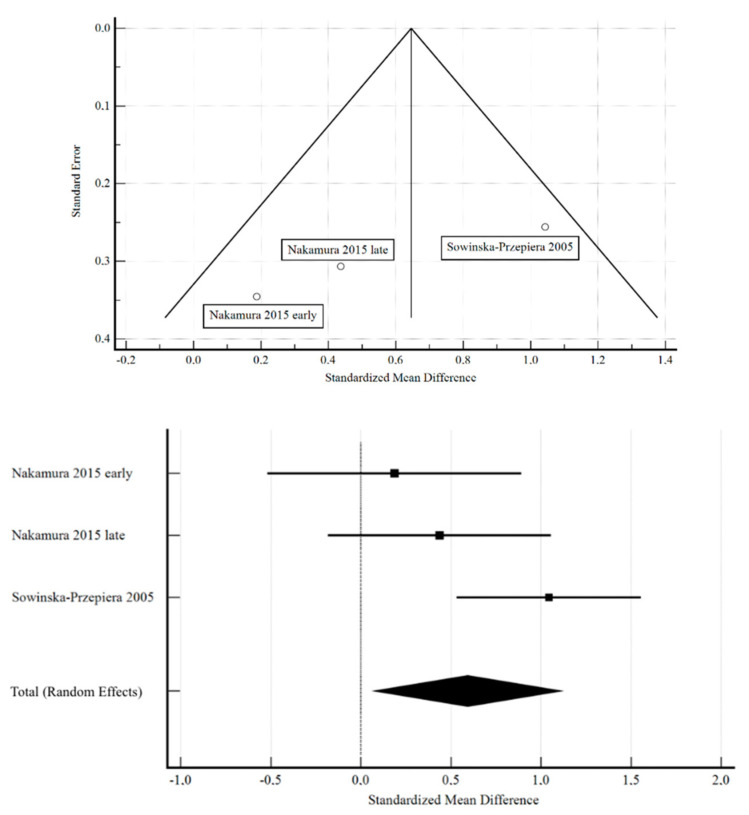
Effect of ERT on BMD [15,16].

**Figure 2 pharmaceuticals-16-01320-f002:**
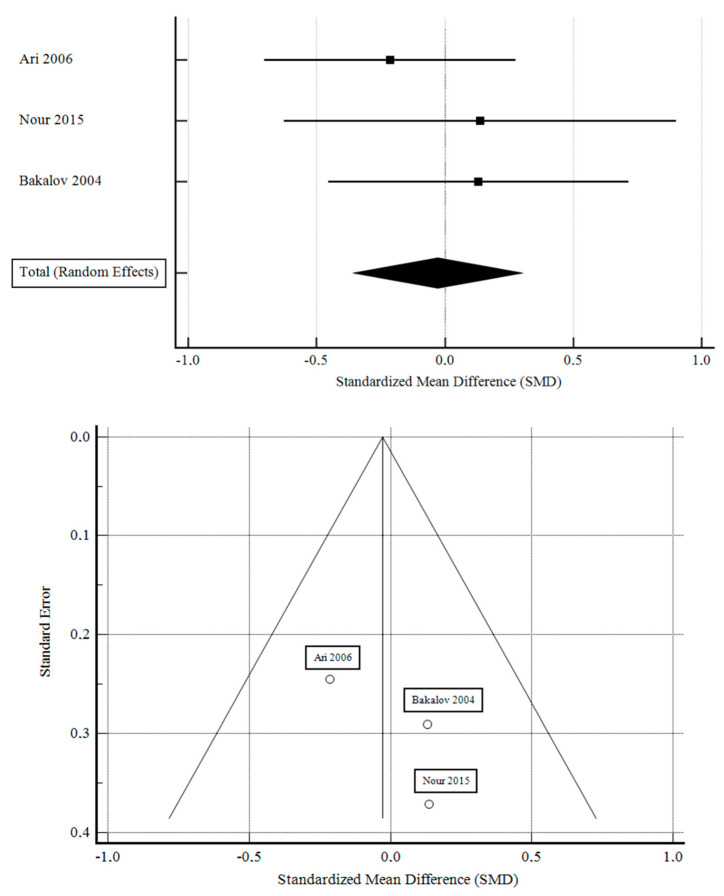
Effect of GHT on BMD [1,17,18].

**Figure 3 pharmaceuticals-16-01320-f003:**
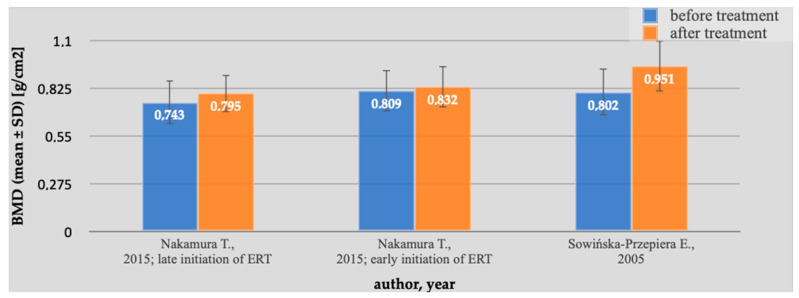
BMD in patients with TS before and after ERT. The studies containing sufficient data on the effect of ERT on BMD demonstrated a significant increase in lumbar spine areal BMD in patients with Turner syndrome. The standardized mean difference in the random model was 0.593 g/cm^2^ (95% CI: 0.0705 to 1.116; *p* = 0.026). The heterogeneity between studies was I^2^ = 56.89% (95% Cl for I^2^ = 0.00 to 87.71; *p* = 0.0983). The results of Egger’s test show that there is a statistically significant publication load. Egger’s test −9.8452 (95% Cl = −29.1591 to 9.2944; *p* = 0.0962) [15,16].

**Figure 4 pharmaceuticals-16-01320-f004:**
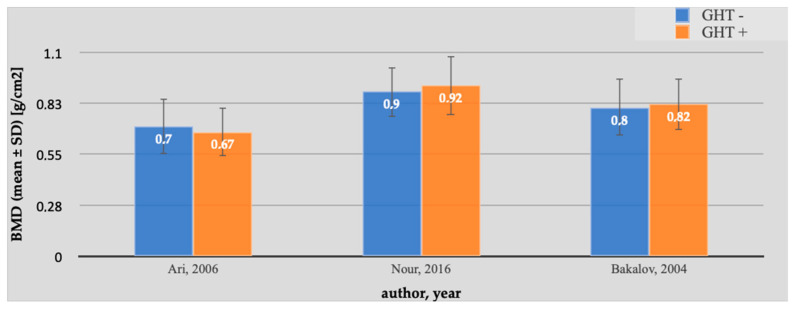
BMD in patients with and without GHT. The studies with sufficient data examining the effect of GHT on BMD demonstrated a non-significant decrease in lumbar spine areal BMD in patients with Turner syndrome. The standardized mean difference in the random model was −0.028 g/cm^2^ (95% CI: −0.359 to 0.303; *p* = 0.868). The heterogeneity between studies was I^2^ = 0.00% (95% Cl for I^2^ = 0.00 to 93.73; *p* = 0.5855). The results of Egger’s test show that there is no statistically significant publication load. Egger’s test 2.9326 (95% Cl = −24.8244 to 30.6897; *p* = 0.4076) [1,17,18].

**Figure 5 pharmaceuticals-16-01320-f005:**
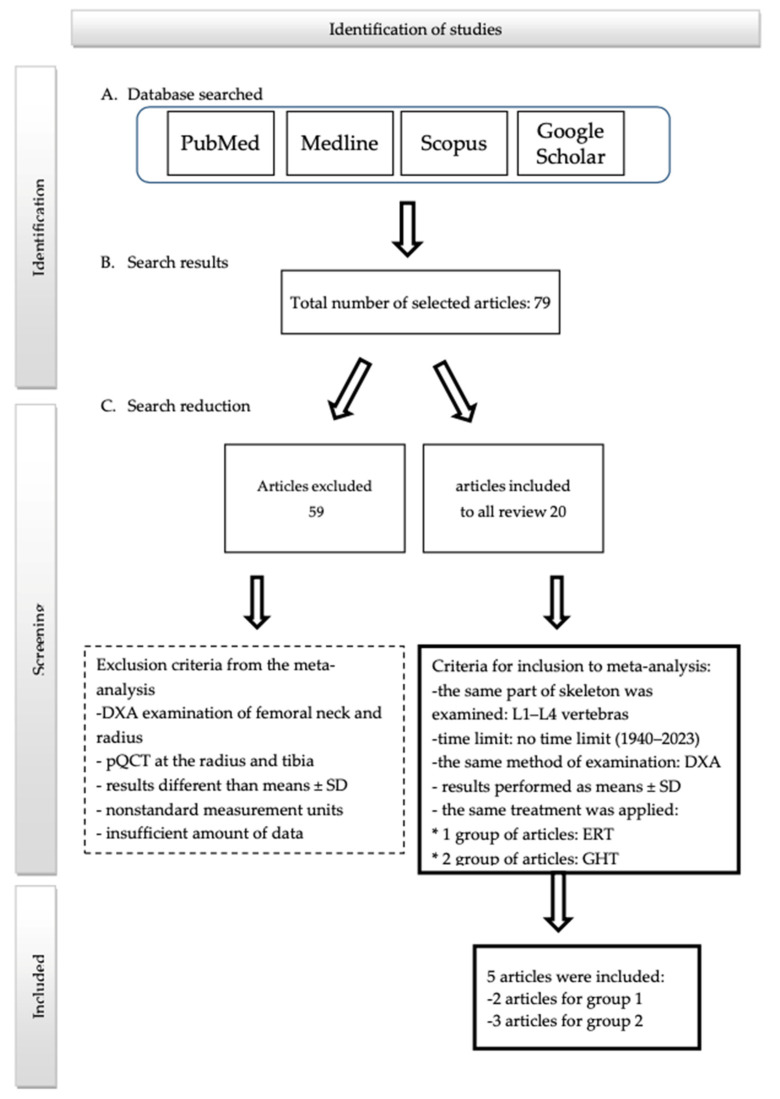
PRISMA Flowchart.

**Figure 6 pharmaceuticals-16-01320-f006:**
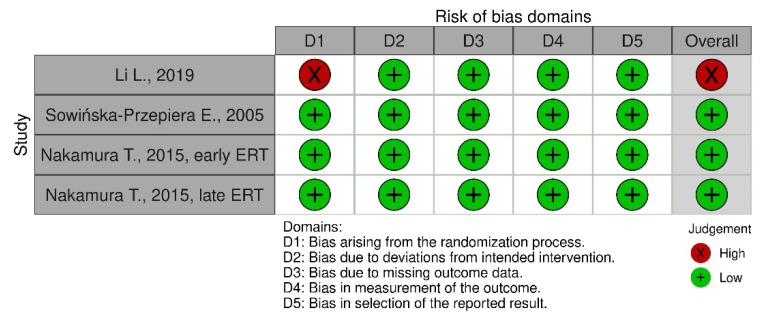
Bias assessment for ERT studies [14,15,16].

**Figure 7 pharmaceuticals-16-01320-f007:**
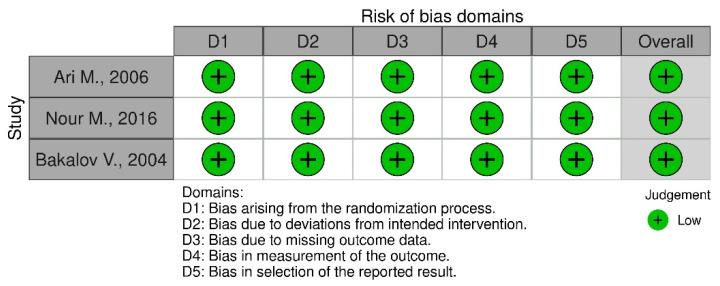
Bias assessment for GHT studies [1,17,18].

**Table 1 pharmaceuticals-16-01320-t001:** Data collected from 3 different sources concerning the amount of the surveyed population, age, GHT, ERT—duration and age of start, BMD before and after treatment and height of girls with TS. The study by Li L. (2019) [14] was excluded from the meta-analysis due to the high risk of bias.

Author, Years	Total Population	Age (Years)	GHT (Number of Group Members)	BMD [g·cm^−3^]		Duration of ERT (Years)	Age of Start ERT (Years)	Height (cm)
				Before ERT	After ERT			
Sowińska-Przepiera E., 2005 [15]	34	22.7 ± 8.2 (12–39)	No treatment	0.802 ± 0.134	0.951 ± 0.148	4 years	22.7 ± 8.2 (12–39)	141.7 ± 8.8
Nakamura T., 2015 [16]	21 (late initiation of ERT)	35.5 ± 8.2 (26–58)	11/21	0.743 ± 0.126	0.795 ± 0.107	13.5 ± 5.7 (7–27)	19.9 ± 1.6 (18–22)	144.7 ± 5.3 (132.0–151.0)
	16 (early initiation of ERT)	28.8 ± 6.1 (18–40)	8/16	0.809 ± 0.120	0.832 ± 0.119	12.8 ± 6.1 (1–23)	16.6 ± 1.2 (14–17)	146.2 ± 5.1 (134.0–151.0)
Li L., 2019 [14]	20	18.45 ± 3 (16–21)	No treatment	0.69 ± 0.09	0.73 ± 0.09	1 year	18.45 ± 3.07 (16–21)	144.89 ± 8.77 (131.8–163)

**Table 2 pharmaceuticals-16-01320-t002:** Data collected from 3 different sources concerning total population, age, ERT, BMD, height and duration of therapy with GH or without this therapy among girls with TS.

Author, Year GHT Application	Total Population		Age (Years)		ERT		BMD [g·cm^−3^]		Duration of GHT (Years)		Height [cm]	
	GHT +	GHT −	GHT +	GHT −	GHT +	GHT −	GHT +	GHT −	GHT +	GHT −	GHT +	GHT −
Ari M., 2006 [1]	39	28	11.9 ± 2.8	12.8 ± 3.1	12/39 (31%)	6/28 (21%)	0.67 ± 0.13	0.70 ± 0.15	4.2 ± 3.2 (1–14)	-----	137 ± 13.6	134 ± 13.9
Bakalov V., 2004 [17]	23	23	21.5 ± 9.4 (7–35)	21.7 ± 9.4 (7–37)	15/16 (94%)	14/16 (89%)	0.82 ± 0.14	0.80 ± 0.16	5.0 ± 2.1	-----	145.2 ± 10.9	143.3 ± 12.3
Nour M., 2016 [18]	12	16	26.7 ± 6.9	28.3 ± 7.2	+	+	0.92 ± 0.16	0.89 ± 0.13	median 5 (2–13)	-----	151.3 ± 6.3	143.9 ± 6.3

## Data Availability

Data sharing is not applicable.

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
