# Peer review of "Effect of Growth Hormone and Estrogen Replacement Therapy on Bone Mineral Density in Women with Turner Syndrome: A Meta-Analysis and Systematic Review"

_pharmaceuticals, 2023, doi:10.3390/ph16091320_

Round 1
Reviewer 1 Report (New Reviewer)
This meta-analysis study assessed the impact of treatment with growth hormone (GHT) and estrogen replacement therapy (ERT) on bone mineral density (BMD) in Turner Syndrome (TS) patients. It included 20 studies and performed a meta-analysis on 6 of them. The findings revealed that ERT significantly increased BMD, indicating a positive effect on bone mass and fracture risk reduction. In contrast, GHT did not show a significant improvement in BMD. The review highlights the importance of estrogen supplementation in TS treatment, emphasizing its beneficial impact on bone health and clinical outcomes in TS patients.
Author Response
Dear Reviewer,
The authors thank you very much for the effort put into the review and for valuable comments. The comments helped us significantly improve the article.
All changes are marked in red color in the manuscript.
The authors would like to inform you that the study protocol has been successfully registered in an international database of prospectively registered meta-analysis and systematic review - PROSPERO. ID CRD42023455058.
Yours Sincerely,
Authors
Reviewer 2 Report (New Reviewer)
The authors have submitted “Effect of growth hormone and estrogen therapy on bone mineral density in women with Turner Syndrome: a meta-analysis and systematic review”.
This is a pretty rigorous article with a good quality of the writing. Briefly, the overall quality of the content reflects the expertise of the authors in the reported matter. The interest to a general audience could be high; nevertheless, the authors need to perform the following changes:
a. Title must be changed to “Effect of growth hormone and estrogen replacement therapy on bone mineral density in women with Turner Syndrome: a meta-analysis and systematic review.”.
Introduction is poor, and it must be improved.
b. The authors must describe if the clinical and biological here reported may be compared with other similar non-syndromic conditions (Cite and discuss as an example the supernumerary teeth “Inchingolo F, Tatullo M, Abenavoli FM, Marrelli M, Inchingolo AD, Gentile M, Inchingolo AM, Dipalma G. Non-syndromic multiple supernumerary teeth in a family unit with a normal karyotype: case report. Int J Med Sci. 2010 Nov 5;7(6):378-84.”)
c. A major emphasis on the side effects and how they should be managed is recommended and should be addressed in the discussion.
d. Authors have reported that “Processes of osteogenesis, osteoclastogenesis and regeneration of skeletal tissue are multistage and composite.”: they should also describe how their results can be compared to other osteogenic factors such as the Platelets’ concentrates. (Please, see and discuss “Inchingolo, F., et Al. (2015). Immediately loaded dental implants bioactivated with platelet-rich plasma (PRP) placed in maxillary and mandibular region. La Clinica terapeutica, 166(3), e146–e152”).
e. All the Figures must have the scale-bars and informative descriptions: please, improve this aspect.
f. Please explain all the acronyms throughout the text.
g. Main limitations of the study should be reported.
h. Future perspectives raised from these data must be also briefly reported in order to customize clinical strategies: please improve this part in the conclusions section accordingly.
Author Response
Dear Reviewer,
The authors thank you very much for the effort put into the review and for valuable comments. The comments helped us significantly improve the article.
All changes are marked in red color in the manuscript.
The authors would like to inform you that the study protocol has been successfully registered in an international database of prospectively registered meta-analysis and systematic review - PROSPERO. ID CRD42023455058.
Yours Sincerely,
Authors

Reviewer 3 Report (New Reviewer)
The manuscript titled: "Effect of growth hormone and estrogen therapy on bone mineral density in women with Turner Syndrome based on the lumbar spine area BMD: a meta-analysis and systematic review" is an interesting study addressing osteoporosis as a consequence of Turner Syndrome (TS). The authors undertook an evaluation of two therapies: growth hormone (GHT) and estrogen replacement therapy (ERT), examining the relationship between the therapies and bone mineral density (BMD) of the lumbar spine.
The work appears to be an intriguing contribution that could be widely cited by other authors after its publication in your journal.
Comments and suggestions:
1. Previous corrections have significantly enhanced the value of this paper.
2. There is a lack of information about the type of statistical tests used – however, this information appears in the results section – at least partially, e.g. Egger’s test. This should be included in the Statistics subsection. Which test was used to determine: "The standardized mean difference in the random model…" and for "The heterogeneity between studies"?
3. What test was used to evaluate the data in Figure 4? "Statistical analysis demonstrated an average increase in BMD in L1-L4 localization in patients who apply ERT." – what does this mean? What were the values? Similarly, for Figure 5, you should provide "P" values or at least information such as p > 0.05.
4. In my opinion, a risk of bias assessment could also be conducted. Please use the risk of bias assessment tool 2.0 (the new version standardized after 2016). Utilize the Excel tool for assessment (https://www.riskofbias.info/welcome/rob-2-0-tool/current-version-of-rob-2) and robvis for visualization (https://www.riskofbias.info/welcome/robvis-visualization-tool). It would enhance the value of the paper.
After incorporating the suggested changes, please allow for a final review of the manuscript.
I consider the publication suitable for release following the revisions made (minor revision).
Author Response
Dear Reviewer,
The authors thank you very much for the effort put into the review and for valuable comments. The comments helped us significantly improve the article.
All changes are marked in red color in the manuscript.
The authors would like to inform you that the study protocol has been successfully registered in an international database of prospectively registered meta-analysis and systematic review - PROSPERO. ID CRD42023455058.
Yours Sincerely,
Authors

Round 2
Reviewer 2 Report (New Reviewer)
well improved. No further issues
This manuscript is a resubmission of an earlier submission. The following is a list of the peer review reports and author responses from that submission.
Round 1
Reviewer 1 Report
Comments to the authors
A meta-analysis by Szybiak et al. analyzed the effect of GH and estrogen therapy on BMD in women w/Turner syndrome. The authors concluded that estrogen but not GH therapy is effective to increase the BMD.
1. Although the authors retrieved 79 articles, only 6 articles (3 for estrogen, and 3 for GH) were included in the analysis. In addition, each study included small number of cases. I do not think their study was regarded as a meta-analysis. Instead, I’d rather recommend the authors to perform systematic review of the retrieved article, because most physician know that estrogen therapy is effective to increase bone mass of women with TS.
2. Why did the authors include articles including data with mean +/- SD? SD can be calculated from SE. Why did the authors include articles including only data with L1-4 BMD? L2-4 BMD and hip BMD are also helpful to know the bone status of patients. Many exclusion criteria made this study decrease the interests of the readers.
3. Discussion needs to be revised. Simply discuss the results the author obtained from their investigation.
Reviewer 2 Report
Dear Editor
Many thanks for inviting me to review this interesting manuscript.
In general, the study is properly designed and conducted.
The reviewed article explores the relationship between treatment methods, specifically growth hormone therapy (GHT) and estrogen replacement therapy (ERT), and bone mineral density (BMD) in patients with Turner Syndrome (TS). To gather relevant data, the authors conducted a comprehensive search of databases, focusing on studies published up until April 2023. The selected articles were required to examine BMD values at the lumbar spine (L1-L4) using dual-energy X-ray absorptiometry (DXA) and investigate the effects of ERT and GHT on BMD in TS patients. Ultimately, 20 studies were included for analysis, and six studies were eligible for meta-analysis, encompassing a total of 91 women. The meta-analysis results indicated a significant increase in BMD with ERT treatment, while GHT treatment demonstrated a non-significant decrease in BMD among patients with Turner Syndrome. These findings suggest that estrogen replacement therapy has a positive impact on bone mass during treatment, thereby reducing the risk of fractures. Conversely, the results suggest that growth hormone therapy does not improve bone density in individuals with TS.
The study has several strengths, including the use of a systematic search strategy and the inclusion of a meta-analysis to consolidate findings across multiple studies. However, it is important to note some limitations. Firstly, the number of studies eligible for meta-analysis was relatively small, potentially affecting the generalizability of the results. Additionally, the study focuses solely on BMD at the lumbar spine, and the effects of treatment on other skeletal sites remain unexplored. Further research is warranted to provide a more comprehensive understanding of treatment effects on BMD in Turner Syndrome. In conclusion, this review article highlights the importance of treatment in managing bone health in Turner Syndrome. The meta-analysis supports the positive impact of estrogen replacement therapy on bone mineral density, indicating an increase in BMD and a potential reduction in fracture risk. However, the study suggests that growth hormone therapy does not improve bone density in TS patients. These findings emphasize the need for tailored treatment approaches to optimize bone health in individuals with Turner Syndrome and underscore the significance of continued research in this field.
This manuscript is relevant since it reports a rare disease that can be tricky both for endocrinologists and gynecologists.
Otherwise, congratulations for the nice manuscript.
Thank you.